# Moderate Seasonal Dynamics Indicate an Important Role for Lysogeny in the Red Sea

**DOI:** 10.3390/microorganisms9061269

**Published:** 2021-06-11

**Authors:** Ruba Abdulrahman Ashy, Curtis A. Suttle, Susana Agustí

**Affiliations:** 1Red Sea Research Center (RSRC), Biological and Environmental Sciences and Engineering Division (BESE), King Abdullah University of Science and Technology (KAUST), Thuwal 23955, Saudi Arabia; ruba.ashy@kaust.edu.sa; 2Department of Biology, College of Science, University of Jeddah, Jeddah 21493, Saudi Arabia; 3Departments of Earth, Ocean and Atmospheric Sciences, and Microbiology and Immunology, The University of British Columbia, Vancouver, BC V6T 1Z4, Canada; suttle@science.ubc.ca; 4Institute for the Oceans and Fisheries, The University of British Columbia, Vancouver, BC V6T 1Z4, Canada

**Keywords:** lysogenic marine microbes, lysogenic marine bacteria, marine viruses, mitomycin C, viral production, lytic, lysogeny, Red Sea, oligotrophic, temperature

## Abstract

Viruses are the most abundant microorganisms in marine environments and viral infections can be either lytic (virulent) or lysogenic (temperate phage) within the host cell. The aim of this study was to quantify viral dynamics (abundance and infection) in the coastal Red Sea, a narrow oligotrophic basin with high surface water temperatures (22–32 °C degrees), high salinity (37.5–41) and continuous high insolation, thus making it a stable and relatively unexplored environment. We quantified viral and environmental changes in the Red Sea (two years) and the occurrence of lysogenic bacteria (induced by mitomycin C) on the second year. Water temperatures ranged from 24.0 to 32.5 °C, and total viral and bacterial abundances ranged from 1.5 to 8.7 × 10^6^ viruses mL^−1^ and 1.9 to 3.2 × 10^5^ bacteria mL^−1^, respectively. On average, 12.24% ± 4.8 (SE) of the prophage bacteria could be induced by mitomycin C, with the highest percentage of 55.8% observed in January 2018 when bacterial abundances were low; whereas no induction was measurable in spring when bacterial abundances were highest. Thus, despite the fact that the Red Sea might be perceived as stable, warm and saline, relatively modest changes in seasonal conditions were associated with large swings in the prevalence of lysogeny.

## 1. Introduction

Viruses are the most abundant microorganisms in oceans [1], varying in number between 10^5^ particles per mL in oligotrophic and deep-sea ecosystems [2,3,4] to 10^8^ particles per mL in productive systems, and comprising an estimated ~10^30^ double-stranded DNA virus particles in the entire ocean. Most of these viruses infect bacteria and are estimated to cause 10 to 50% of the total bacterial mortality in surface waters; thereby killing about 10 to 20% of living biomass in the oceans each day [5,6], and driving the mobilization of nutrients through the viral shunt [7].

Viruses infecting bacteria are known as temperate or lytic phages depending on whether they replicate via a lysogenic or lytic cycle. In lysogeny, the phage DNA integrates into the genome of its host as a prophage and propagates each time the host genome is replicated until an environmental signal triggers the phage to enter the lytic pathway [1,8]. However, some phages, such as the temperate myovirus P1, which infects strains of gram-negative hosts (i.e., *Escherichia coli*) may exist as plasmids and do not integrate into the host chromosome. Myovirus P1 together with its sister P7 can transduce genes between *E. coli* and *Shigella* strains and other plasmid DNAs [9,10]. In the lytic cycle, phage particles are produced resulting in the destruction of the host cell and the release of virions [8,11,12]. The relative importance of lytic and lysogenic pathways for viral replication varies widely among environments [13], and although numerous explanations have been proposed for the cause of this variation, it remains a subject of continuing debate [11,12,14,15,16,17].

Pragmatically, the percentage of lysogenic cells in a sample is typically determined from the proportion of the prokaryotic community that produces phage when exposed to an inducing agent such as mitomycin C, UV radiation or high nutrient concentrations [15]. Temporal changes, such as low nutrients, or low cell biomass are often associated with more lysogeny [18]. The prevalence of lysogeny in oligotrophic systems is related to low host densities as a survival strategy of viruses [19] although data based on virus to microbe ratios abundance suggest that lysogeny is considered to be less dominant in oligotrophic waters [8] and temperate dynamics become increasingly significant at high hosts densities [20]. However, virus-host dynamics could vary in response to seasonal environmental changes, adding variability to a linear response predicted by an empirical relationship.

The Red Sea is a tropical narrow oceanic basin characterized by warm, salty, oligotrophic, and transparent waters [21]. The concentration of nutrients in the Red Sea is low, although shows a latitudinal gradient, with lower concentration in the north areas than in the south [22,23]. The Red Sea is classified as one of the warmest sea basins experiencing rapid warming [23,24]. As well as its tropical location, temperatures vary seasonally between 22 and 32 °C as they are influenced by the monsoon seasons [21]. Reference [25] reported that the high temperature between 31 and 34 °C of the Red Sea was responsible for raising the rates of zooplankton mortality. We tested recently the role of water temperature on viral dynamics and the shift from lysogeny to lytic replication cycles in the Red Sea in a mesotrophic coastal lagoon in the Red Se [26]. The coastal lagoon is a shallow environment separated from the open waters by the reef system, where seawater temperature reached high values (≥33.3 °C) and where we observed that lysogeny occurred in a small percentage at the end of the summer when the temperature was highest and host was low [26].

Although several studies have been performed on microbial communities and bacteria [27,28,29], studies on viral abundances and dynamics in the Red sea are still limited, addressing communities in coral reefs [30,31] and most recently from shoreline waters of the central Red Sea [32], a coastal lagoon [26] and the coastal waters of the Gulf of Aqaba [33], with all studies indicating complex seasonal patterns of abundance and activity of marine viruses.

Hence, the aim of this study was to quantify planktonic viral abundances and dynamics seasonally in an oligotrophic coastal Red Sea location, and to investigate changes in the proportion of lysogenic heterotrophic bacteria and the shift from lytic to lysogenic phases. We hypothesized that lysogeny would be prevalent when heterotrophic bacterial abundance decreased and that high abundance may encompass a reduction of lytic infections as suggested for other oligotrophic systems and will vary with seasonal changes. The approach used to obtain the hypothesis was to test the effect on prophage induction by a chemical treatment with mitomycin C, the most broadly used method in the literature.

## 2. Materials and Methods

### 2.1. Sampling

Time series sampling was done every two weeks from October 2016 to October 2018. Seawater samples were collected using a Niskin bottle from the surface (1 m) of the coastal central Red Sea at a pelagic station away from the reef system (22°30.932′ N, 38°99.739′ E) (Figure 1). Sea surface water temperature (°C) and salinity (PSU) were measured using an Ocean Seven 305 Plus CTD device (Idronaut, Brugherio, Italy) for five to 15 min. The samples were transported to the lab within an hour of collection in refrigerated conditions and preserved form sun. About 300 mL of seawater was filtered through a Whatman glass microfiber GF/F filter (Sigma-Aldrich, Taufkirchen, Germany) for the quantification of chlorophyll-*a* (Chl-*a*) concentration. The filters were kept frozen until analysis (−20 °C). Pigments were extracted using 90% acetone for 24 h and left overnight in the dark at 4 °C. Chl-*a* concentration was estimated with the non-acidification technique using a Turner Design Trilogy Fluorometer, previously calibrated with pure Chl-*a*. Surface seawater was transferred to a 5 L polypropylene container, pre-sterilized with 4% HCl, and pre-rinsed with seawater collected at the same time and place as the samples for the analysis of microbial communities, viral production (VP), and lysogeny. The seawater samples were transported to the lab and duplicated subsamples (1.5 mL each) were taken for the enumeration of viruses and heterotrophic bacteria (HB) by flow cytometry (FCM). From December 2017 to October 2018, the remaining water was sequentially filtered through 20-µm and 2-µm pore-size membranes, and used for the different incubations and preparation of virus-free water (VFW) using a cartridge with a 30 kDa molecular mass cutoff (Millipore).

### 2.2. Virus-Free Water

The virus reduction technique [34,35,36] was performed to measure VP rates and lysogenized bacterial cells. Briefly, the aim of this approach is to maintain the host community while reducing viral abundances [35]. The pre-filtered seawater samples were filtered through a 0.2-µm pore-size membrane (described below) and then ultra-filtrated through a cartridge with a 30,000 Daltons molecular-weight cutoff to obtain 500–600 mL of VFW [35,37].

### 2.3. Incubations for Measurements of Viral Production (Lytic) and Induction from Lysogenic Phase

HB were concentrated by passing three liters of the 20-µm and 2-µm (MF-Millipore filters) pre-filtered seawater through a 0.2-µm-pore-size polycarbonate filter [37] using an Amicon^®^ Stirred Ultrafiltration Cell (8050 Millipore 50 mL–Merck Millipore, Burlington, MA, USA) until a volume of 50–70 mL remained. The concentrated HB cells were then rinsed with 100 mL of VFW five to six times before re-suspending in 250–300 mL of VFW to reduce the concentration of free virus particles [15,37,38].

Six acid-rinsed borosilicate glass flasks were used for the incubation experiments and were filled with 40–50 mL of the washed HB in VFW. Three replicates were amended with 1.0 µg of mitomycin C mL^−1^ [17,37] for lysogen induction and the other three replicates were kept as untreated controls, and the change in viral and bacterial abundances were followed. Mitomycin C is a chemical compound, which inhibits DNA replication and activates DNA repair mechanisms, which causes the RecA protein to cleave a repressor and induces prophages to enter the lytic phase [37]. Mitomycin C is an effective inducing agent for many marine lysogens and this method is widely used, and therefore the results obtained can be compared to those of other studies [37,39]. The flasks were incubated for 24 h in the dark [17,34,39] at the in-situ water temperature at the time of sampling, which varied seasonally from 25.0 to 32.0 °C. Every three hours from the time 0 to 24 h (T_0_ to T_9_), 2 mL was taken from each flask and fixed with 80 µL (1% final concentration) of 25% glutaraldehyde [40] and stored at −80 °C until the analysis of viral and bacterial counts. For the overnight sampling, an Inc-FC204 fraction collector (Gilson, Middleton, WI, USA) equipped with a multiple head and peristaltic pumps was programmed to automatically transfer the samples from the flasks to the cryovials every three hours. Eight cryovials (2 mL) with 80 µL of 25% glutaraldehyde were placed (without lids) in specially fabricated racks that were surrounded by expanded polystyrene and dry ice to keep the samples cold.

We tested the efficiency of fixing and storing the samples in the fraction collector by comparing the abundances of viruses and HB in seawater samples collected during routine sampling (as indicated above) and automatically. At the lab, three replicates of seawater samples were fixed with 25% glutaraldehyde and stored at −80 °C after 10–15 min of fixation. A second set of three replicates was added to the cryovials in the fraction collector rack simulating the conditions of the samples that remained for longer time in the automatic sampling device, i.e., fixed and kept in the dry ice for six hours, and then stored at −80 °C. The comparison was replicated by performing the test in two different sampling events. After two days of storage at −80 °C, heterotrophic bacterial and viral concentrations from all the tested samples were examined by FCM. Automated and manual methods yielded non-significant differences in estimates of viruses abundances (*p* > 0.05, Table 1).

#### 2.3.1. Viral Production Rates

VP rates were estimated following [34] in calculating the rates of VP incubation experiments from the untreated samples. Lytic viral production (LVP) was evaluated from the slope of the linear relationship of the viral abundance increase versus time in the incubation in the first six hours to reduce the new infections occurring during the incubation. As we expected, all viral production assumed during the incubation was as a result of infections prior to the incubation [41].

The formula of the linear regression for calculating LVP used as follows:(1)$$Y=\left(a+b\right)\times X$$
where *Y* is viral abundance per mL during incubation, *a* is the constant, *b* is the slope, and *X* is the time during incubation.

#### 2.3.2. Identification of Viral and Heterotrophic Bacterial Populations by Flow Cytometry (FCM)

Samples for quantification of viral and HB populations were analyzed using a BD FACSCantoTM II flow cytometer (© 2021, Becton, Dickinson and Company, Franklin Lakes, NJ, USA). A series of simplified protocols to identify and enumerate virus populations followed those of [40,42,43], with some modifications as follows: The samples were removed from storage (−80 °C) and thawed at room temperature under sterile conditions. Control samples were prepared by adding 50 µL of an autoclaved 0.2-µm pre-filtered seawater sample to 475 µL of Tris–EDTA buffer (TE, 10 mM Tris and 1 mM EDTA, pH = 8) and 5 µL of SYBR Green I (1:2000) in order to obtain an event rate between 300 and 700 events s^−1^ [40]. The experimental samples were then prepared as for the control samples but adding 50 µL of the time-series samples instead of autoclaved seawater. The tubes were then heated for 10 min in an 80 °C water bath followed by cooling for 5 min in the dark at room temperature. The FCM flow rate was determined before and after running the samples by adding 1 mL of autoclaved Milli-Q containing a total concentration of 10^4^ fluorescent beads and counting for 60 s at the low flow rate (min. 20 mg–max. 60 mg; mean ± (SE), 40 ± 14 mg). Viral populations were identified based on green fluorescence (GF) versus side scatter (SSC). HB were analyzed and prepared following the protocol of [43] by adding 400 µL of each sample stained with 4 µL of SYBR Green I (1:1000) and keeping the samples in the dark for 10–20 min before counting by flow cytometry. The FCM threshold was set up using RF versus GF. Data were recorded and saved in the BD FACSCanto^TM^ II and then analyzed by quantifying viral and bacterial abundances using FlowJo (Version 10.1-Tree star. Inc, Becton, Dickinson and Company, Franklin Lakes, NJ, USA) as described previously [44].

#### 2.3.3. Burst Size and the Percentage of Lysogenic Heterotrophic Bacteria

Control samples of natural seawater (no virus reduction) were incubated in parallel to the viral reduction incubations for the calculation of the burst size (BS) (i.e., the number of phages produced per infected bacterium). The BS was assessed from viruses produced during five of the incubations. BS was then calculated by subtracting the produced virus (VPc) in the untreated control samples from the number of produced virus in the virus-reduced samples (VPr), which demonstrates the net increase in the number of phages that were released from infected bacteria, and then dividing by the number of bacteria killed (Bdead) during the incubations by infection [26], as follows:BS = (VP_c_ − VP_r_)/(B_dead_)(2)
where B_dead_ was calculated as (HB_r_ − HB_c_) the difference between the maximum HB observed in the reduction control and the maximum HB in the natural control where viruses were not reduced.

The percentage of inducible lysogenic HB (%) was calculated following [37]:% Lysogenic HB = (*V_mc_* − *V_c_*)/*BS*/*HBA* × 100(3)
where *V_mc_* is the maximum mean viral abundance of three replicates observed in the mitomycin C-treated samples during incubation, and *V_c_* is the maximum mean of three replicates of viral abundance in the control of standardized values. *BS* and *HBA* refer to the burst size and bacterial abundance at the onset of each experiment, respectively. We also calculated the %Lysogenic HB following a second method [14] using the following equation: %Lysogenic = (B_C_ − B_I_)/B_C_·100, where B_C_ and B_I_ are the number of bacteria enumerated in the control and induced samples at 24 h, respectively. This was corrected by subtracting the average mortality caused by mitomycin C in uninduced samples in all the incubations.

#### 2.3.4. Data Analysis

Statistical analyses were performed using the JMP PRO 14 software (JMP^®^, Version *<14>* SAS Institute Inc., Cary, NC, USA, ©1989–2019). The bivariate test was utilized with the flow cytometry normalized data to determine linear regression between abundances of viruses and bacteria and environmental parameters, and the significance (*p*-value) of this correlation, which is defined at *p* < 0.05.

## 3. Results

### 3.1. Environmental Parameters and Viral and Heterotrophic Bacterial Abundances

Mean monthly records of all the sampling events were retrieved from multiple time series sampling during the study (Figure 2).

The water temperatures during the sampling period fell within 24.0 to 32.5 °C and averaged 29.2 ± 0.33 °C, with minimum values in February and maximum in September for both years (Figure 2A). Chl-*a* concentrations ranged from 0.2 to 0.5 µg L^−1^, with the highest concentration in February and the lowest in July (Figure 2B). Viral abundances ranged from 1.5 to 8.7 × 10^6^ particles mL^−1^ (3.6 × 10^6^ ± 5.5 × 10^5^ particles mL^−1^, mean ± SE), with an increase in February, a peak in October, and low values during spring, summer and the beginning of the winter period (Figure 2C, Appendix A). HB abundances ranged from 1.9 to 3.2 × 10^5^ cells mL^−1^ (2.6 × 10^5^ ± 3.1 × 10^3^ cells mL^−1^, mean ± SE), with increased values in April and at the end of summer during August and September (Figure 2D, Appendix A). The linear regression analyses showed that there were no significant relationships among measured parameters, including viral abundances and water temperature (r^2^ = 0.0049, *p* > 0.05), HB abundances and water temperature (r^2^ = 0.0270, *p* > 0.05), viral and HB abundances (r^2^ = 0.0144, *p* > 0.05), or between the abundances of viruses or HB with Chl-*a* concentration (r^2^ = 0.0082, *p* > 0.05; r^2^ = 0.0010, *p* > 0.05, respectively).

### 3.2. Percentage of Inducible Lysogens and Viral Production

After viral reduction, the time at which viral abundances in both mitomycin-*C*-treated and untreated controls were highest during the 24 h incubations varied seasonally (Figure 3).

Viral abundances after the addition of mitomycin C increased after three to six hours during incubations in January, February, and September (Figure 3), while they peaked after nine hours in July, and 18 h in December and March (Figure 3). In fact, viral abundance in the January incubations in the mitomycin-C-treated samples was higher than in the controls for most of the incubation (Figure 3). Differences in HB abundances also occurred between mitomycin-*C*-treated and untreated samples. HB abundance increased in the untreated controls, but rarely in mitomycin-C-treated samples (Figure 4), where showed a linear decrease with time in some incubations (e.g., April and September, Figure 4e,j) or more often an inflexion in the tendency after 9–12 h of incubation (Figure 4c,d,f–h). Differences in HB abundance between treated and the untreated control were high in the incubations from January, February, April, June and September (Figure 4b,c,e,g,j) although for others were small as observed in May (Figure 4f) and October (Figure 4k). The estimated average burst size (*n* = 5) was 14 ± 6.0 phages per lysed HB and varied between 0.8 in winter and 35.1 in March (Table 2).

The percentage of lysogens varied from undetectable to 13.8% in spring through summer with the highest percentages in winter time, with a maximum of 55.8% in January (Table 2). The % lysogens calculated using the second method based on treated and untrated HB data showed close percentages showing the highest lysogenic bacteria in January with lack of detected lysogens in the same incubations

Temporal changes in HB abundances followed a different pattern to that of inducible lysogens, and was highest in spring when lysogens were lowest, and was relatively low when lysogeny peaked in winter; however, both HB abundance and lysogeny were relatively low in fall (Figure 5). In summer, when temperature was highest, there were modest peaks in HB abundance and inducible lysogens.

Viral production rates were estimated from the initial slope of changes in viral abundances in samples not treated with mitomycin C, but in which the abundance of viruses was reduced. Viral production rates varied highly during the study from negative values to a maximum of 8.97 × 10^5^ virus mL^−1^ h^−1^ in October (Table 2) when viral abundance was highest (Figure 2C). Contrary to lysogeny, the winter had lower lytic viral production, while higher values occurred in spring and fall (Table 2). During summer, viral production was low (Table 2), consistent with low viral and bacterial concentrations (Figure 2 and Figure 5).

## 4. Discussion

This study provides seasonal data on viral abundance and inducible lysogens in the surface waters of the Red Sea, an understudied oligotrophic tropical environment characterized by high temperature and salinity. Most notably, ~56% of the HB contained inducible lysogens in January when HB abundance decreased and were at low abundances compared to their highest abundances in spring, where lysogenic heterotrophic bacterial proportion were undetectable, consistent with a switch from lysogeny to lytic viral production when host abundance is highest. Our results confirm our hypothesis that lysogeny could be an essential mechanism for viral replication in the oligotrophic Red Sea, although lysogeny was not the prevalent phage phase in this study.

Previous studies on viral abundances in the Red Sea waters were limited to examining deep-sea anoxic brines [45] and seasonal dynamics in both shallow waters [32] and a coastal lagoon [26]. Seasonal variability of HB, however, was examined by [29] in the semi-enclosed harbour close to our study location, reporting close numbers of bacterial abundances to those found in our study. Here, we show that abundances of viruses are similar to those from the central Red Sea [32] and the eastern Mediterranean Sea [36,40] as well as other oligotrophic tropical and subtropical waters [46,47,48]. The maximum values were however lower than those found in a mesotrophic coastal lagoon in the Red Sea [26].

We found no significant correlation among viruses, HB, water temperatures, and chl-*a*, like [41], who found that during two years of sampling in the oligotrophic coastal NW Mediterranean, viral abundances were not significantly correlated with other measured parameters. This is in contrast with our results [26] for a coastal Red Sea lagoon waters, where we found significant relationships between viral abundance and bacterial numbers, chl-*a* concentration, and virus-to-bacterium ratio (VBR), and between HB abundance and water temperature. However, the authors found that there was no significant relationship between viral numbers and water temperature. However, the lower range of variability in viruses and trophic degree found in the coastal waters compared to those observed in the lagoon (chl-*a* concentrations varied from 0.5 to 5.1 μg L^−1^, [26] explained the lack of significance. Our results showed a peak of viral abundance in October, at similar period to the peak observed in the coastal lagoon [26]. Both peaks in viral abundance did not follow bacterial peaks, as also observed in the Mediterranean Sea [41]. Moreover, bacterial abundances did not follow a clear seasonal pattern in our study and did not follow peaks in chl-*a* as was also determined in the Red Sea coastal lagoon waters [26]. As discussed by [41], this could be due to viruses contributed from other organisms in the coastal area or because phages could be at different phases of cell infection at the sampling moment. In some studies, peaks in viruses tended to follow peaks in chl-*a* and HB, although this is typically in environments where the dynamic ranges are much larger than in the Red Sea [17,40,47,49,50]. [33] examined taxonomic viral contigs in the Red Sea and found that the seasonal viral response is facilitated by a stable bank population that is readily activated upon increase in abundance of a potential host.

We estimated an average LVP rate of 2.8 × 10^5^ viruses mL^−1^ h^−1^ ± 9.3 × 10^4^ (mean ± SE), which ranged from undetectable in January and May to a peak production rate of 9.0 × 10^5^ viruses mL^−1^ h^−1^ in October when viral abundance was greatest. These results are consistent with higher rates of lytic replication cycles when elevated abundances of viruses and hosts result in higher virus-host contact rates, increased mortality, and higher viral production [16]. LVP rates in this study were close to rates observed in a coastal lagoon Red Sea waters [26], which averaged 2.6 × 10^5^ viruses mL^−1^ h^−1^ (± 6.0 × 10^4^ SE). In contrast, values of LVP in the Red Sea were low compared to those in temperate coastal waters of the NE Pacific [34] and NW Atlantic [50], where viral production rates ranged from 10^6^ to 10^7^ viruses mL^−1^ h^−1^. This is to be expected given the higher productivity and biomass of these environments. In contrast, rates in the Red Sea were comparable to those determined by [51] for the oligotrophic Sargasso Sea and the North Atlantic (0.2 × 10^5^ to 3 × 10^5^ mL^−1^ h^−1^).

Over eleven months we used the viral reduction technique and the chemical inducing agent, mitomycin C [37,49,52] to examine lysogeny in marine microbial communities from the Red Sea. The lowest percentages of lysogenic HB were detected in incubations when decreases in bacterial abundances were not evident from the addition of mitomycin-C, suggesting a lack of phage induction. When prophage induction did occur, it was often not until three to nine hours or even 18 h after mitomycin C addition. This is in contrast to our study in the nearby coastal lagoon [26], where the influence of prophage occurred earlier between three to 12 h of incubation. Besides, [45] reported phage induction after four to eight hours of incubation, with viral and bacterial abundances decreasing in samples incubated for longer than eight hours, whereas we did not observe inhibitory effects on viral and HB abundances after eight hours, and even found phage induction after 18 h. This is consistent with [53], who reported that the largest increase in total virus abundances in mitomycin-C-treated compare to untreated controls was between six and 12 h of incubation, whereas the largest increase in infectious cyanophages was between 12 and 18 h.

The estimated BS in this study was 14 ± 6.0 viruses per bacterium. This value was similar to the estimated BS of 15 ± 5.3 for the coastal lagoon in the Red Sea [26] and compares with values reported for the oligotrophic Gulf of Mexico of 18.92 [54] and 15 to 54 [37], the Sargasso Sea and North Atlantic of ~12 [51], and an average burst size of 19.8 for numerous estimates from other oligotrophic marine environments [55]. Our results show that lysogenic phage production is potentially a significant contributor to total phage production in the coastal Red Sea. The percentage of lysogenic HB was 12.2% ± 4.8 (SE) and ranged from undetectable to 55.8% of HB, which showed a higher proportion of lysogenic phage production than those in the mesotrophic coastal Red Sea that averaging 7.2% ± 2.9 (SE), with the highest percentage of 29.1% determined in October when the bacterial numbers were relatively low (2.7 × 10^5^ cells mL^−1^) and the water temperature was up to 32 °C [26,37] found lower percentages of lysogens in the coastal waters of the Gulf of Mexico ranging from 0.07 to 4.4% of the total bacterial community, and thus most viral infections were lytic rather than lysogenic. Similarly, [56] found little evidence of lysogenic cells in coastal waters of Southern California. Our results indicated that lysogeny was most prevalent during winter and the beginning of spring (March), as well as sporadically in the summer (July and September), contrary to the results presented in the coastal lagoon waters [26], where lysogenic HB did not occur in the winter and in the summer (July). However, the smallest percentage of 2.8% detected during the beginning of spring (March), with a slight occurrence observed in August (late summer) and September (onset of fall), and a peak of 29.1% predominated in the fall (October). About half of the incubation experiments during the eleven months demonstrated lysogens; moreover, the relative abundance of inducible lysogens tended to be highest when lytic viral production was lowest. Other studies have demonstrated the significance of lysogeny induction to bacterial mortality or phage production with lysogens comprising 0 to 41% of the bacteria in Tampa Bay [39]. Likewise, [57] detected that up to 84% of bacteria in the Mediterranean and Baltic Seas were inducible lysogens, with the highest percentages found in deeper waters (800–2000 m).

According to the relationship between lysogenic HB percentage and HB abundances, it can be explained that lysogeny was most prevalent in winter at the time of reduced bacterial abundance, and no detection observed in spring when bacterial abundance peaked. Our results and those reported in our previous study in the Red Sea lagoon [26] indicated undetectable lysogenic bacteria in the spring during times of increased host abundance. This is consistent with other studies showing a negative relationship between lysogeny and higher bacterial production [16,17,26,58]. Hence, we detected the switch from lytic, which was potentially the dominant, to lysogenic phase is related to low host abundance as found by [26,59].

During spring and summer in the Southern Ocean [17], and in the coastal mesotrophic Red Sea [26] when viral and HB abundances are highest, lysogeny was the lowest. Similarly, [16] found that in the Arctic Ocean during summer, there was a much lower proportion of lysogeny when bacterial abundance and production was high, while in the Amundsen Gulf there was a transition from lysogenic to lytic infection as the waters warmed from spring to summer. Likewise, in the Red Sea, lysogeny is greatest in the winter and becomes undetectable by spring as water temperature gradually rises. Although [38,60] suggested that nutrient concentration is not adequate to induce lytic phase in all prophages, [61,62] reported that changes in trophic and temperature conditions may trigger the induction of lysogenic cycle. In this study, we used different in situ temperatures for each incubation experiment, and our results showed that seasonal temperature did not play an essential role in influencing lysogeny. Similar to [45], who found that induction of marine lysogens occurred at elevated in situ temperature in only two out of six experiments performed. The second calculation following a second method described by [14] confirmed the same results. Hence, mitomycin C with the virus reduction technique in this study was found to be a superior method to the other inducing agents [41]. Mitomycin C has been shown to be effective also at much lower concentrations [63,64]. Although HB abundance tended to be lowest in the treated samples, the differences with the abundance in untreated incubations varied highly due to differences in the relevance of lytic and lysogenic phases, and also by the possible presence of different bacterial strains in the community that could vary in their responses to the mutagenic agent [45].

The overall impact of lysogeny on ecosystems remains less clear than the effects of lytic infections. Although prophage can remain stable in the host genome for many generations, they can be triggered to enter the lytic cycle by a variety of chemical, physical, and other inducing agents. Lysogeny may benefit viruses by allowing them to survive periods when host abundance, and thus virus-host contact rates, are low [53]. In September, when water temperature was highest, there was a small increase in lysogeny that was associated with a decrease in HB abundance, suggesting some temperate phage sought refuge in the host genome. Here it can wait for conditions to improve, while its DNA is maintained by the host. In turn, the virus may benefit the host by protecting it against infection by related lytic phages, carry beneficial genes such as CRISPR arrays and metabolic genes, and modulate host metabolism [11,15,65,66]. We observed the alteration from the lysogenic to lytic phase as the season moved from winter to spring, in conjunction with known inducing factors such as increased solar radiation, higher growth rates, and warming temperatures. Although there is an increasing understanding of the mechanisms affecting lysogeny in the marine milieu, much remains to be learned on the impact of lysogeny on bacterial populations and marine ecosystems, and the factors determining the balance between lytic and lysogenic infection [11,14,41,53].

We observed that phage lytic infection dominated in these oligotrophic waters in agreement with the Piggyback-the-Winner model [20] wherein temperate dynamics become increasingly important at high microbial densities. However, we identified high lysogens during lower host’s abundance, which in agreement with [33] highlights that microbial communities in this ecosystem can experience high dynamics at short time scales in response to environmental changes. Our knowledge of viral dynamics and lysogeny in the world’s oceans remains poorly studied, as is evidenced by the few data from the Red Sea [26]. Here we show that viruses are part of a dynamic microbial system in which inducible lysogens varied from undetectable to more than half of the population in winter. As mitomycin C cannot induce all viable prophage, this remains an underestimate of the total proportion of lysogens in the community [67]. Different bacterial strains in the natural marine community vary in their responses to various mutagenic agents, which could be either killed before the influence occurs or be induced by those induction agents [45]. More experiments including testing different concentrations of mitomycin C and other mutagenic agents are required [63,64]. Although lysogeny was not the dominant mechanism of phage infection throughout the year in these oligotrophic waters, it was important in winter when HB abundance and productivity are lower than in the spring where no detection of lysogeny. Thus, despite the fact that the Red Sea might be perceived as stable, warm and saline, relatively modest changes in seasonal conditions were associated with large swings in the prevalence of lysogeny. This emphasizes the need to develop a much more in-depth understanding of the factors regulating lysogeny in natural bacterial communities, and the processes driving microbial population dynamics.

In summary, our study extended previous findings to demonstrate that changes in viral and HB populations, and lysogeny, are highly dynamic in the warm and saline waters of the oligotrophic Red Sea. Our results indicated that lytic phage infection dominated but suggest that lysogeny could be also an important mechanism for viral replication in the oligotrophic Red Sea. We detected lysogeny induction in the stressful summer conditions when maximum temperature and high oligotrophy persisted, however, the maximum percentage of lysogens were detected in the winter when HB abundance decreased. Further studies are required to identify natural inducing agents and the role of lysogeny in determining the abundance and genetic diversity of marine microbial communities.

## Figures and Tables

**Figure 1 microorganisms-09-01269-f001:**
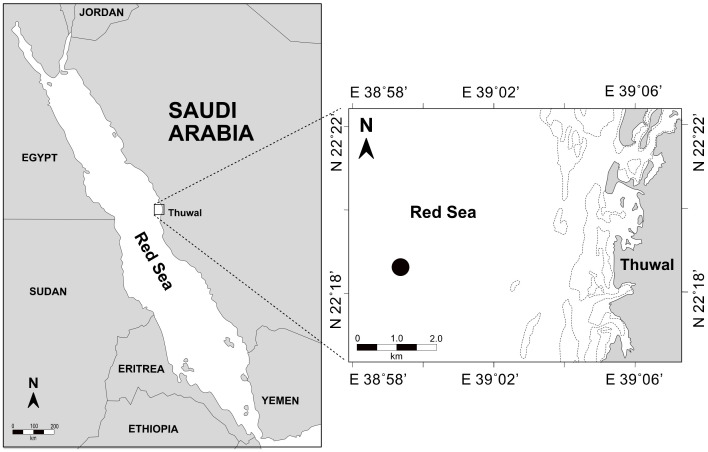
Location of the sampling station in the central coastal Red Sea (Saudi Arabia) where samples were collected from 2016 to 2018.

**Figure 2 microorganisms-09-01269-f002:**
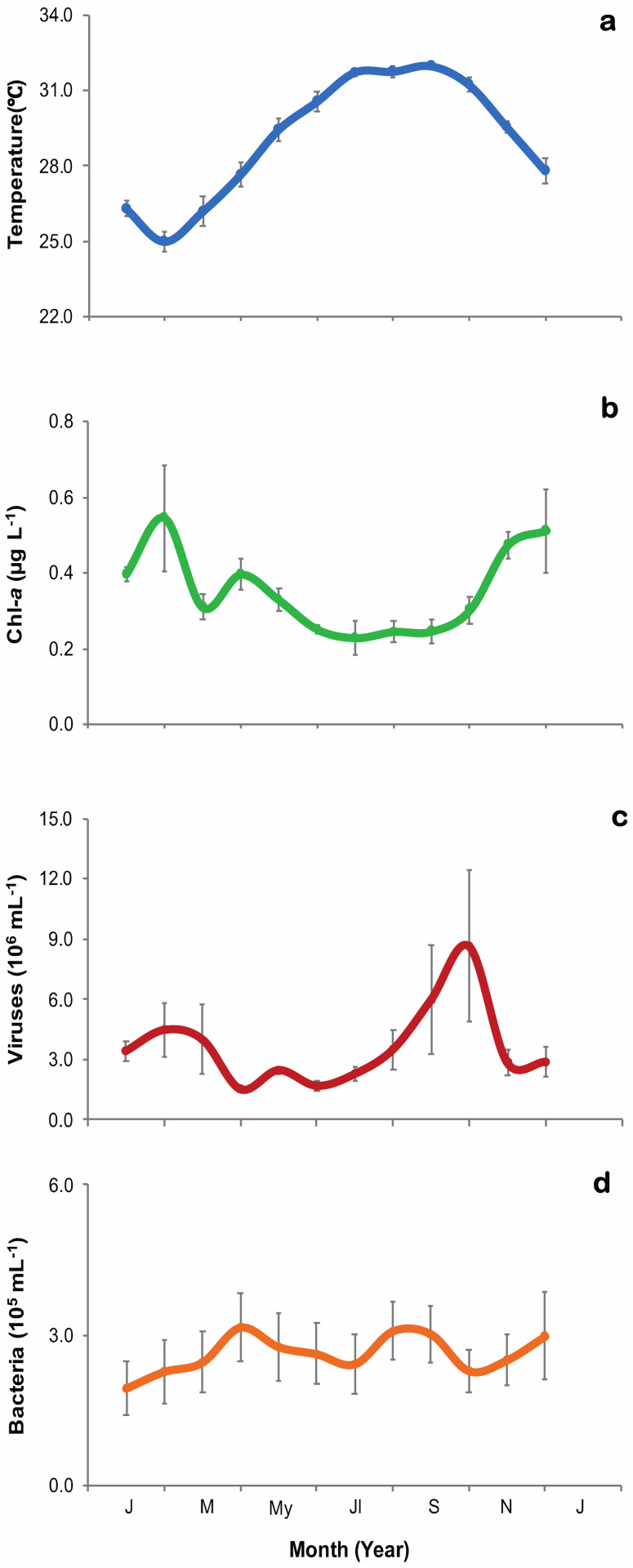
Monthly averaged (±SE) environmental parameters during the time-series sampling in the Red Sea from all the sampling events from 2016 to 2018. (**a**) Surface water temperature (blue line and dots), (**b**) Chl-*a* concentration (green line and dots), (**c**) Viral abundance (red line and dots), and (**d**) heterotrophic bacterial abundance (orange line and dots).

**Figure 3 microorganisms-09-01269-f003:**
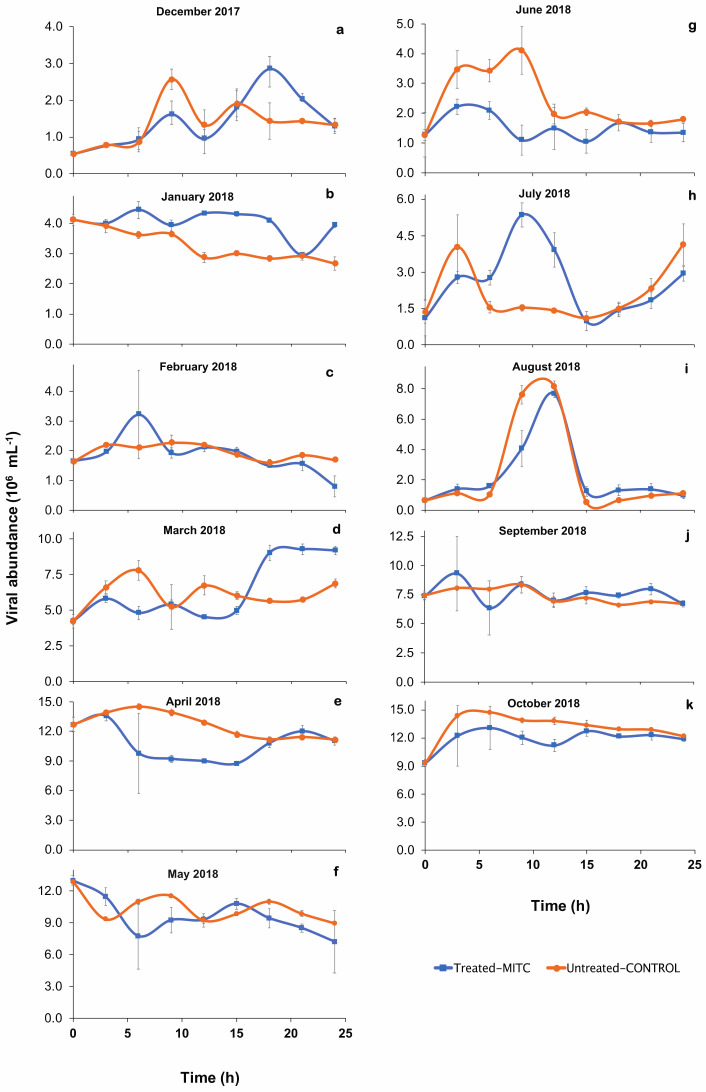
Viral abundance during the monthly 24-h incubations performed for the induction of lysogens. The plots (**a**–**k**) represent different months of induction. Error bars are standard errors of three replicates for each incubation time.

**Figure 4 microorganisms-09-01269-f004:**
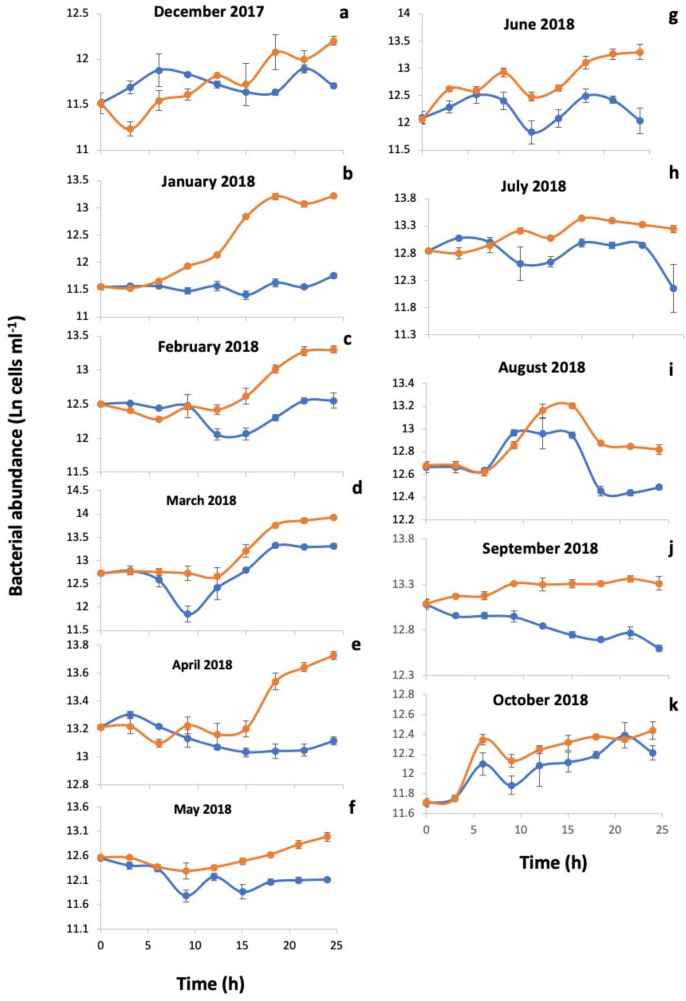
Changes in heterotrophic bacterial abundance with time in incubations treated with mitomycin C and in untreated controls. The plots (**a**–**k**) represent different months of induction. Blue lines and dots correspond to samples treated with mitomycin C (1 µg mL^−1^), while orange lines and dots correspond to the untreated controls. Error bars are standard errors of three replicates for each incubation time.

**Figure 5 microorganisms-09-01269-f005:**
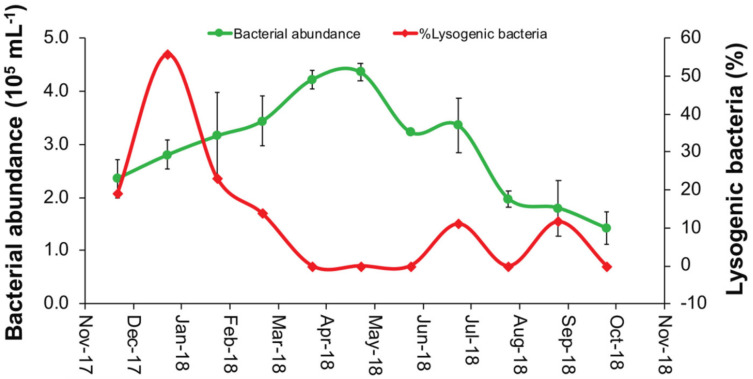
Percentages of lysogenic heterotrophic bacteria (red line and diamonds) and monthly mean (±SE) bacterial abundance (green line and dots) quantified in the natural microbial communities of the coastal Red Sea during the second year of the study (December 2017 to October 2018).

**Table 1 microorganisms-09-01269-t001:** Comparison of the viral and heterotrophic bacterial (HB) abundances (mean ± SE) obtained using manual or automated sampling. There were not significantly differences between methods (*t*-test, *p* > 0.05) ^1^.

Sample	Viral Abundance (10^6^ mL^−1^) (Manual)	Viral Abundance (10^6^ mL^−1^) (Automatic)	HB Abundance (10^5^ mL^−1^) (Manual)	HB Abundance (10^5^ mL^−1^) (Automatic)
Sample (1)	6.3 ± 0.1	5.5 ± 0.8	3.5 ± 0.09	3.3 ± 0.2
Sample (2)	6.9 ± 0.4	6.5 ± 0.2	5.1 ± 0.1	4.9 ± 0.1

^1^ Viruses: *t*-test sample 1, *t* = 1.017, df = 9, *p* = 0.822; *t*-test sample 2, *t* = 0.897, df = 9, *p* = 0.784). HB: *t*-test sample 1, *t* = 0.469, df = 9, *p* = 0.673; *t*-test sample 2, *t* = 0.977, df = 9, *p* = 0.819.

**Table 2 microorganisms-09-01269-t002:** Estimation of percentages of lysogenic heterotrophic bacteria (HB) and Lytic Viral Production (LVP), ± standard error of the replicates, based on induction by mitomycin C for monthly individual incubation experiments since December 2017 to October 2018.

Sampling	LVP (mL^−1^ h^−1^)	Lysogenic HB (%)	Lysogenic HB * (%)	Burst Size (Phage Bacteria^−1^)
December-17	2.1 × 10^5^ ± 4.7 × 10^4^	19.2 ± 20.5	14.7	N/A ^2^
January-18	N/D ^1^	55.8 ± 8	66.5	N/A
February-18	6.0 × 10^4^ ± 2.7 × 10^4^	23.1 ± 8.7	39.4	0.8 ± 2.3
March-18	6.0 × 10^5^ ± 1.2 × 10^5^	13.8 ± 1.7	4.2	35.1 ± 22
April-18	3.1 × 10^5^ ± 1.7 × 105	N/D	N/D	N/A
May-18	N/D	N/D	N/D	3.9 ± 2
June-18	2.8 × 10^5^ ± 4.0 × 10^5^	N/D	N/D	N/A
July-18	3.2 × 10^4^ ± 3.4 × 10^4^	11.1 ± 5.1	11.0	4.9 ± 1.7
August-18	7.0 × 10^5^ ± 1.8 × 10^4^	N/D	N/D	25.2 ± 7.6
September-18	9.0 × 10^4^ ± 1.4 × 10^5^	11.7 ± 1.3	34.5	N/A
October-18	9.0 × 10^5^ ± 5.0 × 10^4^	N/D	N/D	N/A

^1^ N/D = Not detectable; ^2^ N/A = not available; * following Williamson et al. 2002.

## Data Availability

Data of virus and bacterial abundances is available in Appendix A. Other data of the pelagic time series is available upon request.

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
