# Peer review of "Moderate Seasonal Dynamics Indicate an Important Role for Lysogeny in the Red Sea"

_microorganisms, 2021, doi:10.3390/microorganisms9061269_

Round 1
Reviewer 1 Report
Review of the manuscript by Ashy et al., entitled “Strong seasonal dynamics indicate an important role for lysogeny in the Red Sea”, submitted to Microorganisms for consideration.
Overall comment: In this manuscript, the Authors present a novel dataset on the quantification of planktonic viral abundances and seasonal (lytic and lysogenic) dynamics in an oligotrophic coastal area of the Red Sea, concluding that relatively modest changes in seasonal conditions can be associated with large shifts in the prevalence of lysogeny. Overall, the article is written quite well, but needs relevant revisions in all sections. I detail here below some comments to improve this manuscript for publication.
Specific comments
L58 and ref 20 should be better acknowledged/discussed, as such relevant line of evidence (and related following papers) suggest a higher relevance of lysogeny with increasing host abundance. That is, a trend opposed to the one evidenced by the Authors in the present manuscript.
L69 and in some other sentences related to ref [26], “they”.. is not so appropriate, as these are virtually the same Authors.. The work performed by Ashy and Agustì, published in Viruses in 2020, should be more evident (also using that info for the abstract.. for comparison) to allow the reader understand that a very similar (complementary) work has been done in another site nearby by the same Authors. Overall, I feel the Authors should better acknowledge and integrate the results from that previous work within this new manuscript. Also in Figure 1, by excluding from the map the nearby North-East lagoon area investigated in their previous published work, the Authors seem to want to “hide” this information. This is not necessary, conversely the point in the lagoon should be highlighted, with related reference in the table legend and comparison of results in the text, when needed/appropriate (I see that the Authors have compared and discussed this in some points, but this can surely be improved and implemented with additional effort).
L81: The approach used to “test such” hypothesis was to “assess”…
L86: biweekly might be misleading: every two weeks or twice a week?
L100:How can the Authors be sure to detect only “bacteria” as written throughout the text (or, “heterotrophic bacterial populations” as in Viruses 2020)? Should this be interpreted/changed to “prokaryotes”?
L111-112 please specify model/producer of the ultrafiltration device
L123-124 Even though few studies exist, the Authors might want to add a sentence to acknowledge that in similar studies, mitomycin C has been shown to be effective also at much lower concentrations. Thus, testing different mitomycin C concentrations in future studies of such kind might be useful to check/reduce biases due to the possible toxic effects of mitomycin C on the host cells (citing for example Rastelli et al., 2017 EnvironMicrobiol; Yoshida-Takashima et al., 2013 Extremophiles).
L128-129: This short sentence should be joined with the previous one.
L148-149: “yielded similar and non-significant estimates of abundances…” The Authors should rephrase this wrong sentence (i.e., like “the samples displayed no significant differences (p>0.05)” or similar)
L150 “p = 0.822 > 0.05” and similar, this is quite redundant… actual p values might be expressed within Table 1, thus simplifying the main text.
L158-161 Not clear how the Authors can be sure of this fact. More likely, they should change “avoid” with “reduce” at L159, and delete the conclusion expressed in L160-161.
L163 Should VP be LVP?
L169. Could the Authors provide parallel (control) data based on filter-based quantification of viruses and cells, and provide comparison/regression with FCM? Otherwise, please discuss something about this.
L194: please report the rationale and related calculation in this manuscript also (i.e., not only by a reference to the previous Viruses 2020 paper). Indeed, the procedure used for BS assessment is not completely clear. Why the “no virus reduction” would serve as a control? How was the factor “Bdead” assessed?
L199 what does “maximum mean” refer to in this calculation? At which time point? Do the Authors assume that the viral abundances were the same in the control and mit-C treated microcosms at the start of the experiment? Or, are these data obtained from independent microcosms at the starting point?
L191-202 I think that the Authors should support these results with additional parallel calculation methods (see for example Williamson et al 2002 ApplEnvMicrob on results from Tampa Bay), also considering possible alternatives for BS determination or estimation. Notably, BS data are “not shown” (L 247) despite these are the base for most of the presented dataset. This aspect needs to be fixed.
L243-245: this passage is not clear, could the Authors better explain? Should this be better explained in the discussion section?
Figure 3 Issue 1. How can the Authors explain the cases/timepoints in which viral abundance values in the mit-C treatments are significantly lower than in the respective non-treated controls? Issue 2. I notice several quick drops in viral counts over time, likely indicating not negligible viral decay.. can the Authors discuss this point related to viral production estimates?
Figure 4 “Examples” of data are not sufficient. All the data on cell counts over time must be reported here to properly complement Figure 3 on viral data. Possibly, the figures can be joined, setting cell abundance data on a secondary y axis.
L259 Figure 3 caption and similar for Figure 4: The term “changes in” might be misleading, as the Figures rather shows the “abundance” at different time points (i.e., over time). Figure 4 L267-268, not needed as already in the color-legend within the Figure
Table 2, the Authors should provide mean values and standard deviations
L283 “this study is contributing”àbetter, “this study provides”?
L284-285 Could the Authors provide reference for this sentence and better explain this fact? By the way, this aspect is more related with the introduction section and the Authors might decide to avoid such part at the beginning of the discussion of their results.
L291-293 This is too general… the data are not sufficient for this wide-scale and general conclusion.
L317 please better explain this “pathogens” hypothesis
L319-320 also this sentence should be better explained, as the concept is not obvious.
L321-323 Ref 33 should be better introduced and discussed in relation with the data presented by the Authors in this manuscript.
L338-339 please also see comment to Figure 3. The meaning of this sentence is not obvious, since in several cases the viral abundances in mit-C treatments are lower than in their respective non-treated controls. Logically, this is also due to the mit-C treatment.
L352-356 please consider the previous comment on BS (L191-202)
L363-378 could the Authors acknowledge more papers/experimental evidences from available literature on % lysogens? I feel that the few cited papers do not provide a sufficient background on this topic.
L391 “the other end of the globe” sounds a bit odd, please rephrase
L421-422 please acknowledge also here the previous work published in Viruses 2020
L426-428 This is related to my comment about mit-C concentrations used in the experiments and related literature that took this aspect into account (see comment to L123-124). Please discuss/provide suggestion for similar future experiments.
L437-446 this final part could be improved and be less fragmented. Please also acknowledge also in this case the previous work in the nearby area. The “stress” concept at L441 sounds not so appropriate in the context of this manuscript, please better rephrase.
Author Response
Reviewer #1. Comments and Suggestions for Authors.
Review of the manuscript by Ashy et al., entitled “Strong seasonal dynamics indicate an important role for lysogeny in the Red Sea”, submitted to Microorganisms for consideration.
Overall comment: In this manuscript, the Authors present a novel dataset on the quantification of planktonic viral abundances and seasonal (lytic and lysogenic) dynamics in an oligotrophic coastal area of the Red Sea, concluding that relatively modest changes in seasonal conditions can be associated with large shifts in the prevalence of lysogeny. Overall, the article is written quite well, but needs relevant revisions in all sections. I detail here below some comments to improve this manuscript for publication.
Authors: We are grateful to the reviewer for the detailed revision helping us to improve the manuscript.
Following suggestions of Reviewer #2, we changed the title as follows: “Moderate seasonal dynamics indicate an important role for lysogeny in the Red Sea”
Specific comments
L58 and ref 20 should be better acknowledged/discussed, as such relevant line of evidence (and related following papers) suggest a higher relevance of lysogeny with increasing host abundance. That is, a trend opposed to the one evidenced by the Authors in the present manuscript.
Authors: We agree and in the revised version we discussed our results in relation to the piggy-back theory.
In the introduction we added in Lines 65XX-66: “However, virus-host dynamics could vary in response to seasonal environmental changes, adding variability to a linear response predicted by an empirical relationship.”
And to the discussion, we added the following statement (lines 644-649): “We observed that phage lytic infection dominated in these oligotrophic waters in agreement with the Piggyback-the-Winner model [20] wherein temperate dynamics become increasingly important at high microbial densities. However, we identified high lysogens during lower host's abundance, which in agreement with [33] highlights that microbial communities in this ecosystem can experience high dynamics at short time scales in response to environmental changes. “
L69 and in some other sentences related to ref [26], “they”.. is not so appropriate, as these are virtually the same Authors. The work performed by Ashy and AgustiÌ€, published in Viruses in 2020, should be more evident (also using that info for the abstract.. for comparison) to allow the reader understand that a very similar (complementary) work has been done in another site nearby by the same Authors. Overall, I feel the Authors should better acknowledge and integrate the results from that previous work within this new manuscript. Also in Figure 1, by excluding from the map the nearby North-East lagoon area investigated in their previous published work, the Authors seem to want to “hide” this information. This is not necessary, conversely the point in the lagoon should be highlighted, with related reference in the table legend and comparison of results in the text, when needed/appropriate (I see that the Authors have compared and discussed this in some points, but this can surely be improved and implemented with additional effort).
Authors: We revised and changed ‘they” as suggested by the reviewer.
The methodology is similar to that of the previous study and the results are complementary. However, the previous work corresponded to a very different ecosystem than the one studied here. The previous study was made in a shallow coastal lagoon, that varied from oligotrophic to mesotrophic throughout the year, although in this study we were sampling open coastal and oligotrophic waters. The proximity of the lagoon is not a relevant aspect as they are different ecosystems separated indeed by the reef systems. So, proximity in Km in the map do not imply the two systems will be physically and ecologically close. Moreover, the complexity of the reef barrier implies long navigation to reach the lagoon, so the sampling was made on different days in the two ecosystems. How can we hide the location of the lagoon when it is published and the previous study is cited?
In any case, we extended the information about the relative proximity of the lagoon and the description of the two ecosystems in the revised version in lines 79-84, we added the paragraph:
“We tested recently the role of water temperature on viral dynamics and the shift from lysogeny to lytic replication cycles in the Red Sea in a mesotrophic coastal lagoon in the Red Se [26]. The coastal lagoon is a shallow environment separated from the open waters by the reef system, where seawater temperature reached high values (³ 33.3 â—¦C) and where we observed that lysogeny occurred in a small percentage at the end of the summer when the temperature was highest and the host was low.”
As indicated below, we also revised the information in the discussion to clarify that the two ecosystems are different.
L81: The approach used to “test such” hypothesis was to “assess”... L86: biweekly might be misleading: every two weeks or twice a week?
Authors: We revised and changed to “every two weeks”.
L100:How can the Authors be sure to detect only “bacteria” as written throughout the text (or, “heterotrophic bacterial populations” as in Viruses 2020)? Should this be interpreted/changed to “prokaryotes”?
Authors: We revised and changed to heterotrophic bacteria; the autotrophic prokaryotes identified in the flow cytometer were not included.
L111-112 please specify model/producer of the ultrafiltration device
Authors: We added the information: “using a cartridge with a 30kDa molecular mass cutoff (Millipore)”.
L123-124 Even though few studies exist, the Authors might want to add a sentence to acknowledge that in similar studies, mitomycin C has been shown to be effective also at much lower concentrations. Thus, testing different mitomycin C concentrations in future studies of such kind might be useful to check/reduce biases due to the possible toxic effects of mitomycin C on the host cells (citing for example Rastelli et al., 2017 EnvironMicrobiol; Yoshida Takashima et al 2013 Extremophiles)
Authors: We added to the methods section the following sentence: “Mitomycin C is an effective inducing agent for many marine lysogens and this method is widely used, and therefore the results obtained can be compared to those of other studies [37-39].”
And in the discussion section, we added the following information in lines 623-624: ”Mitomycin C has been shown to be effective also at much lower concentrations Yoshida Takashima et al 2013-[63], Rastelli et al., 2017-[64]
L128-129: This short sentence should be joined with the previous one.
Authors: We reorganized as indicated above.
L148-149: “yielded similar and non-significant estimates of abundances...” The Authors should rephrase this wrong sentence (i.e., like “the samples displayed no significant differences (p>0.05)” or similar)
Authors: We changed the sentence.
L150 “p = 0.822 > 0.05” and similar, this is quite redundant... actual p values might be expressed within Table 1, thus simplifying the main text.
Authors: We agree and removed.
L158-161 Not clear how the Authors can be sure of this fact. More likely, they should change “avoid” with “reduce” at L159 and delete the conclusion expressed in L160- 161.
Authors: Changed
L163 Should VP be LVP?
Authors: We changed to LVP
L169. Could the Authors provide parallel (control) data based on filter-based quantification of viruses and cells, and provide comparison/regression with FCM? Otherwise, please discuss something about this.
Authors: We did not make a parallel quantification.
L194: please report the rationale and related calculation in this manuscript also (i.e., not only by a reference to the previous Viruses 2020 paper). Indeed, the procedure used for BS assessment is not completely clear. Why the “no virus reduction” would serve as a control? How was the factor “Bdead” assessed?
Authors: We added the information in lines 269-282: “Control samples of natural seawater (no virus reduction) were incubated in parallel to the viral reduction incubations for the calculation of the burst size (BS) (i.e., the number of phages produced per infected bacterium). The BS was assessed from viruses produced during five of the incubations. BS was then calculated by subtracting the produced virus (VPc) in the untreated control samples from the number of produced virus in the virus-reduced samples (VPr), which demonstrates the net increase in the number of phages that were released from infected bacteria, and then dividing by the number of bacteria killed (Bdead) during the incubations by infection [26], as follows:
BS = (VPc − VPr)/ (Bdead) (2)
Where Bdead was calculated as (HBr – HBc) the difference between the maximum HB observed in the reduction control and the maximum HB in the natural control where viruses were not reduced.”
L199 what does “maximum mean” refer to in this calculation? At which time point? Do the Authors assume that the viral abundances were the same in the control and mit-C treated microcosms at the start of the experiment? Or, are these data obtained from independent microcosms at the starting point?
Authors: We clarified this aspect in the revised manuscript: “Where Vmc is the maximum mean viral abundance of three replicates observed in the mitomycin C-treated samples during incubation, and Vc is the maximum mean of three replicates of viral abundance in the control of standardized values.”
L191-202 I think that the Authors should support these results with additional parallel calculation methods (see for example Williamson et al 2002 ApplEnvMicrob on results from Tampa Bay), also considering possible alternatives for BS determination or estimation. Notably, BS data are “not shown” (L 247) despite these are the base for most of the presented dataset. This aspect needs to be fixed.
Authors: We added the BS values to Table 2 in the revised manuscript as requested. We indeed applied the second method of Williamson et al. 2002, and our results were in agreement with the data presented. We added these new results to Table 2 for comparison.
L243-245: this passage is not clear, could the Authors better explain? Should this be better explained in the discussion section?
Authors: We modified the sentence and moved to the discussion as indicated by the reviewer.
Figure 3 Issue 1. How can the Authors explain the cases/timepoints in which viral abundance values in the mit-C treatments are significantly lower than in the respective non-treated controls? Issue 2. I notice several quick drops in viral counts over time, likely indicating not negligible viral decay.. can the Authors discuss this point related to viral production estimates?
Authors: We clarified this aspect.
Figure 4 “Examples” of data are not sufficient. All the data on cell counts overtime must be reported here to properly complement Figure 3 on viral data. Possibly, the figures can be joined, setting cell abundance data on a secondary y axis.
Authors: We built a new Figure 4, including the HB data on cell counts for all the incubations. We represented independently as joining to the plots showing virus abundance in Fig. 3 was noisy. A new description of the results was added to the revised manuscript: “ Differences in HB abundances also occurred between mitomycin-C-treated and untreated samples. HB abundance increased in the untreated controls, but rarely in mitomycin-C-treated samples (Figure 4), where showed a linear decrease with time in some incubations (e.g. April and September, Fig. 4E, J) or more often an inflexion in the tendency after 9-12 h of incubation (Fig. 4 C, D, F, G, H). Differences in HB abundance between treated and the untreated control were high in the incubations from January, February, April, June and September (Figure 4 B, C, E, G, J) although for others were small as observed in May (Fig. 4F) and October (Fig. 4K).”
L259 Figure 3 caption and similar for Figure 4: The term “changes in” might be misleading, as the Figures rather shows the “abundance” at different time points (i.e., over time). Figure 4 L267-268, not needed as already in the color-legend within the Figure
Authors: We corrected.
Table 2, the Authors should provide mean values and standard deviations L283 “this study is contributing”aÌ€better, “this study provides”?
Authors: We corrected as indicated by the reviewer and added the standard errors to the means of Table 2.
L284-285 Could the Authors provide reference for this sentence and better explain this fact? By the way, this aspect is more related with the introduction section and the Authors might decide to avoid such part at the beginning of the discussion of their results.
Authors: We agree and removed from the paragraph.
L291-293 This is too general... the data are not sufficient for this wide-scale and general conclusion.
Authors: We removed the sentence.
L317 please better explain this “pathogens” hypothesis
Authors: We corrected in the revised version.
L338-339 please also see comment to Figure 3. The meaning of this sentence is not obvious, since in several cases the viral abundances in mit-C treatments are lower than in their respective non-treated controls. Logically, this is also due to the mit-C treatment.
Authors: We removed the sentence.
L352-356 please consider the previous comment on BS (L191-202)
Authors: As indicated above, we added the data to Table 2; also calculated % lysogeny following Williamson et al. 2002, showing consistent results.
L363-378 could the Authors acknowledge more papers/experimental evidences from available literature on % lysogens? I feel that the few cited papers do not provide a sufficient background on this topic.
Authors: In this paragraph, citations are focused on temporal studies and ecosystems close to the one studied here.
L391 “the other end of the globe” sounds a bit odd, please rephrase
Authors: Changed.
L421-422 please acknowledge also here the previous work published in Viruses 2020
Authors: Done
L426-428 This is related to my comment about mit-C concentrations used in the experiments and related literature that took this aspect into account (see comment to L123-124). Please discuss/provide suggestion for similar future experiments.
Authors: We added this aspect as suggested by the reviewer in lines 666-667: “More experiments including testing different concentrations of mitomycin C and other mutagenic agents are required [63, 64].”
L437-446 this final part could be improved and be less fragmented. Please also acknowledge also in this case the previous work in the nearby area. The “stress” concept at L441 sounds not so appropriate in the context of this manuscript, please better rephrase.
Authors: The last paragraph was modified as follows: “ In summary, our study extended previous findings to demonstrate that changes in viral and HB populations, and lysogeny, are highly dynamic in the warm and saline waters of the oligotrophic Red Sea. Our results indicated that lytic phage infection dominated but suggest that lysogeny could be also an important mechanism for viral replication in the oligotrophic Red Sea. We detected lysogeny induction in the stressful summer conditions when maximum temperature and high oligotrophy persisted, however, the maximum percentage of lysogens were detected in the winter when HB abundance decreased. Further studies are required to identify natural inducing agents and the role of lysogeny in determining the abundance and genetic diversity of marine microbial communities.”
Reviewer 2 Report
The manuscript by Ashy et al. provides a seasonal account on the viral life strategies and highlights the importance of lysogeny in a tropical marine environment (Red Sea). The authors have done a good job in collecting large amount of data (two years) in a stable and relatively unexplored environment. This study is of interest to aquatic microbial ecologists as it provides baseline data and information on viral strategies in such ecosystem. In spite of the above, the authors have not clearly revealed the environmental factors contributing to percent lysogeny except the fact that low bacterial abundances coincided to high percentage lysogeny. More detailed statistical analyses can help, as the authors have collected large volume of data.
Specific comments
Title: Need to modify the title: The authors conclude relatively modest changes in seasonal conditions in the studied environment but the title tells otherwise.
L16: modify it as “….viral dynamics (abundance and infection)…”
L16: with can be replaced by “characterized by”
L17: should be 22°-32° C, pls. provide salinity in brackets
L17: change to “thus making it a stable….”
L19: two years; indicate years in brackets
L26-27: Which factor(s) led to large swings for percent lysogeny?
L77-83: when the authors refer to viral dynamics equal importance should also be given to viral lytic infection apart from lysogeny. This part need to be revised.
L96: How were surface water samples from 1m depth collected?
L99: In what time samples were transported to laboratory after collection? Precautions taken while transporting? Whether they were well protected from heat and light radiations?
L102: can you specify the nature of filters (20µm and 2µm)?
L215: chl a was nearly stable (0.2 – 0.5µg/L) during the studied period. Therefore, addressing them as largest peak and lowest values does not hold good.
L218: here and elsewhere. In such a tropical system, I don’t think the sampling period needs to be differentiated into spring, summer, autumn and winter. This may hold better for temperate systems given the wide variations in water temperature.
L221-226: no need to show the p values when relationships are not significant
Figure 2: Sampling events 2016 to 2108 does not correspond. I see only one full year. Its not clear?
L236-239: viral abundances highest at different incubation times? What does it say.
L274-281: Can the lytic viral production can be converted to % of lysed bacteria or mortality? I think the authors can do it and then compare with percent lysogeny. Do the authors observe any relation between lytic infection and lysogeny?
L289: bacterial productivity? Did the authors measure this?
L365: something is missing. Please correct this sentence.
The discussion is too long. Discussion of results in relation to other marine and oceanic systems could be reduced or shortened.
Author Response
Reviewer #2. Comments and Suggestions for Authors
The manuscript by Ashy et al. provides a seasonal account on the viral life strategies and highlights the importance of lysogeny in a tropical marine environment (Red Sea). The authors have done a good job in collecting large amount of data (two years) in a stable and relatively unexplored environment. This study is of interest to aquatic microbial ecologists as it provides baseline data and information on viral strategies in such ecosystem. In spite of the above, the authors have not clearly revealed the environmental factors contributing to percent lysogeny except the fact that low bacterial abundances coincided to high percentage lysogeny. More detailed statistical analyses can help, as the authors have collected large volume of data.
Authors: We are grateful to the reviewer for the helpful revision and comments on the manuscript.
Specific comments
Title: Need to modify the title: The authors conclude relatively modest changes in seasonal conditions in the studied environment but the title tells otherwise.
Authors: We agree and changed the title as follows: “ Moderate seasonal dynamics indicate an important role for lysogeny in the Red Sea “.
L16: modify it as “....viral dynamics (abundance and infection)...” L16: with can be replaced by “characterized by”
L17: should be 22°-32° C, pls. provide salinity in brackets
L17: change to “thus making it a stable....”
Authors: We corrected.
L19: two years; indicate years in brackets
L26-27: Which factor(s) led to large swings for percent lysogeny?
Authors: We revised , but we are not sure about the comment on the factors, as we could not add too many words without exceeding the number of words.
L77-83: when the authors refer to viral dynamics equal importance should also be given to viral lytic infection apart from lysogeny. This part needs to be revised.
Authors: We revised the paragraph, and now reads: “to investigate changes in the proportion of lysogenic heterotrophic bacteria and the shift from lytic to lysogenic phases. We hypothesized that lysogeny would be prevalent when heterotrophic bacterial abundance decreased and that high abundance may encompass a reduction of lytic infections as suggested for other oligotrophic systems and will vary with seasonal changes.”
L96: How were surface water samples from 1m depth collected?
Authors: We added the information to the methods: “using a Niskin bottle”
L99: In what time samples were transported to laboratory after collection? Precautions taken while transporting? Whether they were well protected from heat and light radiations?
Authors: It was within an hour as the boats used are fast and the location is not far from the University harbour; moreover, samples were transported refrigerated and preserved from the sun. We clarified in the revised manuscript.
L102: can you specify the nature of filters (20μm and 2μm)?
Authors: Added (MF Millipore).
L215: chl a was nearly stable (0.2 – 0.5μg/L) during the studied period. Therefore, addressing them as largest peak and lowest values does not hold good.
Authors: Corrected
L218: here and elsewhere. In such a tropical system, I don’t think the sampling period needs to be differentiated into spring, summer, autumn and winter. This may hold better for temperate systems given the wide variations in water temperature.
Authors: We revised to reduce emphasis and removed it from Figure 5.
Figure 2: Sampling events 2016 to 2108 does not correspond. I see only one full year. Its not clear?
Authors: We clarified and the Figure legend now reads: “ Monthly averaged (±SE) environmental parameters during the time-series sampling in the Red Sea from all the sampling events from 2016 to 2018. “
L236-239: viral abundances highest at different incubation times? What does it say.
Authors: We corrected
L274-281: Can the lytic viral production can be converted to % of lysed bacteria or mortality? I think the authors can do it and then compare with percent lysogeny. Do the authors observe any relation between lytic infection and lysogeny?
Authors: We presume it could be but we did not find the correction; it was not significant.
L289: bacterial productivity? Did the authors measure this? L365: something is missing. Please correct this sentence.
Authors: We corrected
The discussion is too long. Discussion of results in relation to other marine and oceanic systems could be reduced or shortened.
Authors: We tried to revise and simplify, although reviewer #1 suggested some additions to the discussion.
Round 2
Reviewer 1 Report
The revisions by the Authors can be considered sufficient to endorse publication.
Reviewer 2 Report
The authors have sufficiently adressed my concerns